# In-Line Rheo-Optical Investigation of the Dispersion of Organoclay in a Polymer Matrix during Twin-Screw Compounding

**DOI:** 10.3390/polym13132128

**Published:** 2021-06-29

**Authors:** Paulo F. Teixeira, José A. Covas, Loïc Hilliou

**Affiliations:** Institute for Polymers and Composites, University of Minho, 4800-058 Guimarães, Portugal; p.teixeira@dep.uminho.pt

**Keywords:** organoclay, polymer nanocomposites, extrusion, dispersion, rheo-optics, in-process monitoring, rheology

## Abstract

The dispersion mechanisms in a clay-based polymer nanocomposite (CPNC) during twin-screw extrusion are studied by in-situ rheo-optical techniques, which relate the CPNC morphology with its viscosity. This methodology avoids the problems associated with post extrusion structural rearrangement. The polydimethylsiloxane (PDMS) matrix, which can be processed at ambient and low temperatures, is used to bypass any issues associated with thermal degradation. Local heating in the first part of the extruder allows testing of the usefulness of low matrix viscosity to enhance polymer intercalation before applying larger stresses for clay dispersion. The comparison of clay particle sizes measured in line with models for the kinetics of particle dispersion indicates that larger screw speeds promote the break-up of clay particles, whereas smaller screw speeds favor the erosion of the clay tactoids. Thus, different levels of clay dispersion are generated, which do not simply relate to a progressively better PDMS intercalation and higher clay exfoliation as screw speed is increased. Reducing the PDMS viscosity in the first mixing zone of the screw facilitates dispersion at lower screw speeds, but a complex interplay between stresses and residence times at larger screw speeds is observed. More importantly, the results underline that the use of larger stresses is inefficient per se in dispersing clay if sufficient time is not given for PDMS to intercalate the clay galleries and thus facilitate tactoid disruption or erosion.

## 1. Introduction

After the demonstration by Vaia and Giannelis [1] that direct polymer melt diffusion into the galleries of organically-modified layered silicates can be achieved, melt mixing of organoclay and polymer (usually by twin-screw extrusion) was adopted as the most popular route for the mass production of clay-based polymer nanocomposites (CPNC) [2,3,4,5,6]. These materials have found increasing practical applications in barrier packaging (films and bottles), in flame-retardant electrical cables, as anticorrosive coatings on metals, in rubber automotive compounds, among many others [7]. Nevertheless, despite the significant body of work on the manufacture of these materials by melt mixing techniques, the delivery of a fully exfoliated CPNC, i.e., a material made of individual nano-sized clay platelets homogeneously dispersed into the polymer matrix, is rarely achieved. Instead, practical polymer—clay nanocomposites comprise structures with sizes ranging from nanometers to micrometers [2,3,4].

A stepwise mechanism was proposed to explain the development of this multiscale structure along the extruder. Clay dispersion in the polymer melt includes the diffusion of the polymer chains within the clay interlayer spacing (intercalation) of the tactoids and the delamination of the individual platelets (exfoliation) [8,9]. Clay modification by exchanging hydrated inorganic cations for organic ones bearing aliphatic chains, which results in the production of so-called organoclay, is central to melt intercalation and the delivery of CPNC for a wide range of applications [2,3,4,5,6,7]. More important than the expansion of the interlayer space, which facilitates the diffusion of polymer chains inside the clay interlayer spacing, clay modification lowers the energy of adhesion between clay lamellae below the energy of polymer chain scission (break-up of C-C covalent bond) [9]. Coming back to the dispersion mechanism, large stresses first promote the disruption of aggregates of clay tactoids into single tactoids. These are subsequently shear-ruptured into smaller platelet stacks, which enables platelet-by-platelet peeling of tactoids under sufficient shear, and polymer diffusion into the clay galleries, eventually leading to clay exfoliation [8,10]. This well-accepted mechanism is supported by a large body of studies where extrusion conditions are varied and related to the level of clay dispersion and resulting CPNC properties. The extensive literature on the topic has been recently reviewed by Vergnes [11], who discussed in detail the interplay between stress, flow rate, residence time, and the extent of clay dispersion. Essentially, most studies point towards the enhancement of melt intercalation and clay platelet exfoliation by larger stresses and longer residence times which, to some extent, relate to the total strain experienced by the CPNC in the extruder. Larger stresses can be generated either by increasing the screw speed or the viscosity of the polymer matrix, whereas the total strain can be varied by tuning the ratio between the feeding rate and the screw speed. Linear relationships between the extent of clay exfoliation (measured by the elasticity of the clay network in the CPNC melt) and the total strain were established [12], whereas the specific mechanical energy of the process was proposed as a quantitative indicator of the morphology achieved by twin-screw extrusion [13].

Nevertheless, the sequential break-up/peeling mechanism can be questioned. For instance, the importance of polymer chain diffusion between clay galleries to facilitate the separation of clay platelets and the subsequent clay exfoliation has been demonstrated [9,14], which highlighted the antagonistic need of low viscosity for melt intercalation and of large viscosity for tactoid break-up and complete platelet exfoliation [14,15]. Moreover, a significant number of reports present results that do not fit in the above dispersion mechanism and/or the conventional extrusion parameters–clay dispersion relationships. For example, larger screw speeds not only did not improve the exfoliation of organo-montmorillonite in a recycled PET matrix, but even reduced the extent of melt intercalation [16]. Increasing the molecular mass of the polymer (i.e., the hydrodynamic stresses) had no effect on the melt intercalation [17]. Contrarily, heating the barrel (thus, lowering the stresses) yielded a better clay dispersion [18]. Furthermore, the leveling-off of clay dispersion or even a reversion of dispersion at larger screw speeds or feeding rates has been reported [11]. It has been shown that the thermal degradation of the organic clay modifier, occurring at larger screw speeds or smaller feeding rates, due to large viscous heating and/or longer residence times, respectively, actually triggers polymer diffusion out of the clay galleries and the eventual collapse of the latter [19]. Degradation has an even more acute impact on clay dispersion when thermally sensitive polymer matrices are used to manufacture biodegradable CPNC. For instance, larger screw speeds are detrimental to the dispersion of clay in poly(lactic acid), as demonstrated recently by the in-line rheo-optical monitoring of twin-screw extrusion [20].

Therefore, the literature suggests that the interplay between process parameters, materials properties, and final levels of clay dispersion is complex. In twin-screw extrusion, stress and residence times are difficult to control independently, both depending on the screw profile, screw speed, feed rate, and barrel temperature, as these dictate the local polymer viscosity, velocity patterns, and degree of channel fill. The strategy used to feed the materials to the extruder impacts equally on those relationships. Most reports rely on the feeding of polymer + clay premixes. This is a practical consequence of the small size of most laboratory extruders, operating with outputs usually lower than 5 kg/h and the consequent difficulty in the precise dosing of low clay incorporation percentages (typically less than 5 wt.%). This procedure differs from the masterbatch dilution or sequential feeding approaches used in industry. Yet, the few studies on the impact of the feed strategy on clay dispersion document controversial results. Masterbatch dilution was reported to be more effective than premixes in exfoliating clay [21,22], whereas a similar extent of clay dispersion was seen using either premix or sequential feeding [18]. A further complexity may be added by the probable thermally-induced degradation of the polymer and/or clay material during many of the experiments reported in the literature, eventually polluting the results presented. Indeed, most studies rely on the post-extrusion characterization of samples. Such characterization often involves subjecting the material to an additional thermal and/or mechanical cycle, which may change the CPNC structure actually developed during extrusion. Similar changes can also take place when sampling materials from opened barrels after stopping and freezing the extrusion run. Therefore, efforts have been made to identify relationships between the degree of dispersion assessed by optical means (e.g., light attenuation) and the rheological properties of CPNC [11,20,23] since both experimental techniques can be performed in-situ during extrusion [20].

It is, therefore, interesting to study the dispersion mechanisms of polymer–clay nanocomposites adopting methodologies that avoid possible thermal degradation effects and post-extrusion characterization of the materials whilst controlling stress and residence time. In order to accomplish this, the present work adopts simultaneously three innovative approaches:-A rheo-optical die is coupled to a small-scale prototype twin screw extruder. The in situ characterization of the CPNC morphology and viscosity bypasses all issues related to a possible post-extrusion structural rearrangement.-A model CPNC, namely an organo-modified montmorillonite in a polydimethylsiloxane (PDMS), is used. This CPNC can be processed at ambient or low temperatures (up to 100 °C, which is well below the onset for PDMS or organo-clay thermal degradation [24]). Indeed, PDMS with relatively low molecular mass has been used in the CPNC literature [25,26] as the rheological study of polymer dynamics in the presence of clay is readily amenable at room temperature. Similarly, PDMS has been used to study flow-induced structures in colloidal suspensions (see, e.g., [27] and references therein).-Processing this CPNC avoids the need to superimpose polymer melting with mixing.

Rheo-optical methods have been used previously both off-line [23] and in-line [20] to study the dispersion of organoclay in poly (lactic acid) (PLA) matrices. PDMS-clay nanocomposites have been previously characterized but were prepared by solution intercalation methods [25,26,28]. To the authors’ knowledge, this is the first time that PDMS-organoclay nanocomposites are prepared by melt mixing and that their dispersion is monitored by in-process rheo-optical methods. Moreover, the incorporation of clays in a melt rather than the joint feeding of solid premixes is rarely reported in the literature.

Keeping constant both clay content and feed rate, the effect of screw speed and partial heating of the barrel on the viscosity curves, average particle radii, and corresponding volume fraction of the nanocomposite are measured in-line, and the usefulness of creating an initially lower viscosity to enhance melt intercalation before the application of larger stresses for tactoids disruption and clay exfoliation is tested.

## 2. Materials and Methods

### 2.1. Materials

The polymer matrix used in this study is a polydimethylsiloxane (PDMS) AK 1,000,000 (Wacker-Chemie GmbH, München, Germany) with a dynamic viscosity of 970 Pa.s at 25 °C and a density of 0.97 g/cm^3^, as reported by the manufacturer who also indicates a thermal stability during hours up to 150 °C. An organo-modified montmorillonite (sodium form) clay (Dellite 72 T, Laviosa Chi. Mine., Livorno, Italy) was used as filler. The modification of this layered silicate with dimethyl dehydrogenated tallow ammonium results in an inter-lamellae distance of 2.85 ± 0.05 nm, as measured on dried powder samples [29]. This clay has a density of 1.7 g/cm^3^, and a powder particle size of 7–9 µm, as reported by the manufacturer. The clay powder was dried overnight at 120 °C in a ventilated oven prior to compounding.

### 2.2. Compounding and In-Process Characterization of CPNC

A prototype modular mini-scale fully intermeshing co-rotating twin-screw extruder was used to disperse the clay powder in PDMS. This equipment, detailed in length elsewhere [29], reproduces the range of shear rates and residence times usually found in industrial environments. Different screw elements can be slid on two shafts in order to create different screw profiles. The assemblage is then inserted in a barrel made of interconnected modules. In the experiments reported here, the screw length *L* over the screw external diameter *D* (13 mm) ratio is *L/D* = 27. Most barrel modules are equipped with a feed port (which can be tapped) and three sampling devices, which enable the swift collection of small amounts of material from the screw channel along the barrel without stopping the process. Figure 1 shows the construction of the barrel, the locations of the sampling ports, as well as the screw configuration adopted for this study (the maximum channel depth is 1.5 mm). The latter includes two mixing zones, each made of a block of kneading disks staggered −60°, which are separated by conveying elements. Since PDMS is a fluid at room temperature, it was fed with a syringe pump (PHD 22/2000, Harvard Apparatus, Holliston, MA, USA) immediately downstream of the location of the first sampling port, whereas clay was fed with a volumetric feeder (Piovan MDP1, S. Maria di Sala, Italy) right after the second sampling location.

The extruder is coupled to a rheo-optical die which was used in earlier studies for the in-line assessment of clay dispersion in poly(lactic acid) [20,29]. The die contains two flow channels for the extrusion of rectangular tapes. Both channels are equipped with valves at the entrance to allow for the independent control of material outputs in each channel, while maintaining constant extruder screw speed and feed rate. One of the channels is fitted with melt pressure transducers (two flush-mounted Dynisco PT422A (0–3000 PSI and 0–750 PSI)), thus enabling the in-line measurement of flow curves (steady shear viscosity *η* at different shear rates $\dot{\gamma }$), as described in length elsewhere [30]). Moreover, optical windows give access to small-angle light scattering (SALS) and turbidity measurements for the assessment of the radius *R* and volume fraction *ϕ* of particles scattering the light (see refs [18,30] for a detailed description of the optical train used for SALS and turbidimetry). The rheo-optical die and optical train for SALS and turbidimetry are schematized in Figure 2.

Thus, the rheo-optical die has the unique capability to relate the CPNC morphology with its rheological characteristics without changing the thermomechanical environment in the extruder. For all experiments, the CPNC output was maintained constant by fixing a feed rate of 156 g/h for the PDMS and 4 g/h for the clay (which corresponds to 2.5 wt.% clay content). The latter is still small enough to avoid significant multiple light scattering, but sufficiently large to achieve rheological and X-ray diffraction sensitivity to the level of clay dispersion. Only two parameters were varied, screws speed and temperature of the two first barrel modules (no heating or heating to 100 °C). The average residence time *t* of the CPNC was measured in transient experiments. While PDMS was being processed at a constant feed rate, clay was added (at time zero). The time until changes in melt pressure or turbidity were seen was then noted as *t*.

### 2.3. Off-Line Characterization of CPNC

The thermo-rheological characterization of PDMS was performed with a stress-controlled rotational rheometer (ARG2, TA Instruments, New Castle, DE, USA) in a parallel plates configuration (40 mm diameter and 0.5 mm gap). Flow curves were measured at different temperatures by sweeping up the shear rates and reporting the steady shear viscosity (considered steady when less than 5% variation in the value recorded during 10 s is achieved) for each shear rate applied for a minimum of 20 s. Wide-angle X-ray diffraction (WAXD) spectra of extruded CPNC were acquired using a Bruker D8 Discover diffractometer (lambda source 0.154 nm). Samples were scanned (2θ range from 1.4 to 5° with steps of 0.04°, in order to track any possible change in the d_001_ basal spacing of the organo-modified clay) within 1 h after CPNC production, in view to limit the effects of morphological relaxation [30,31] on measured peaks intensities and widths.

## 3. Results and Discussion

### 3.1. Rheological Characterization of the Polymer Matrix

Flow curves of PDMS measured at different temperatures are presented in Figure 3. The polymer behaves essentially as a Newtonian fluid in the range of shear rates tested with the parallel plates, with the onset for shear-thinning being delayed to larger shear rates with increasing temperature (see Figure 3a). The temperature dependence of the zero shear viscosity *η*_0_ follows an Arrhenius behavior, as suggested by the Arrhenius fit to the data reported in the inset to Figure 3a. This fit returns an activation energy *E* = (13.9 ± 0.8) kJ/mol, which is in harmony with the values documented in the literature [32]. The flow curves measured in-line during two extrusion runs without heating the barrel are plotted in Figure 3b. They are highly reproducible and show that PDMS is shear thinning at the shear rates developing in the extruder for the screw configuration and speeds tested [20].

The Cross equation for shear-thinning fluids was used as a constitutive equation to fit the shear rate ($\dot{\gamma }$) dependence of the shear viscosity *η* plotted in Figure 3b due to the small size of the data set (15 points), and to the easy extraction of the zero shear viscosity *η*_0_ from the data [33]:(1)$$\eta =\frac{\eta_{0}}{1+{\left(\tau \dot{\gamma }\right)}^{n}}$$
where *τ* is the relaxation time and *n* the shear-thinning exponent. Note that the limiting viscosity at an infinite shear rate, originally proposed in Cross formulation [33], is taken as nil in Equation (1) as such limiting value does not show up in Figure 3b. The fit to the data (see Figure 3b) returns *η*_0_ = 838 ± 22 Pa.s, *τ* = 0.025 ± 0.002 s and *n* = 0.68 ± 0.04 Pa.s. A comparison with *η*_0_ values reported in Figure 3a suggests that the PDMS temperature in the measuring slit channel is 31 °C. This is nearly 7 °C above the actual temperature of the PDMS measured at different locations along the barrel and at the die exit. The mismatch in *η*_0_ is thus to be assigned to the inherent precision of the in-line and off-line rheological measurements. Indeed, a precision of 10% for slit rheometry and rotational rheometry is generally admitted (see [34] and references therein).

### 3.2. Effect of Screw Speed

The flow curves of the CPNC measured in-line during extrusion at four different screw speeds and with no heating of the barrel, are displayed in Figure 4. These flow curves lie above the one of the polymer matrix as expected from the reinforcing effect of clay [26,31]. Only a few studies report the flow properties of CPNC at large shear rates relevant for melt processing (i.e., in excess of 10 s^−1^). Capillary rheometry of injection molded samples of CPNC with a Nylon 6 matrix showed that the flow curves of CPNC remained below those of the polymer matrix at larger shear rates [10]. Such reduced viscosity related to a larger shear thinning in the CPNC was explained by the flow-induced orientation of nanoscale platelets. In another report though, the same group also called upon nylon degradation and slip at the clay—polymer interface to explain the drop in CPNC shear viscosity [2]. The SALS pattern pictured in the inset to Figure 4 is isotropic, thereby ruling out any flow-induced particle orientation in the measuring slit channel. Note, however, that the cut-off length scale for SALS sensitivity is of the order of 100 nm for the experimental configuration used here, and thus the orientation of smaller clay objects under steady shear [35] cannot be assessed. Measurements of CPNC temperature at the exit of the rheo-optical die returned a value of 24 ± 1 °C for all tested screw speeds, which is essentially the ambient and die temperature. As such, viscous heating can be ruled out to explain the drop in the flow curves in Figure 4.

Flow curves at lower shear rates are shifted downwards as the screw speed increases, whereas data seem to converge at the higher shear rate range. Therefore, increasing stress and reducing the residence time in the extruder has an impact on both the zero shear viscosity and the shear thinning behavior. The trend reported in Figure 4 seems to be at odds with the larger yield stresses and shear thinning exponents commonly reported for the linear rheological response of CPNC processed at larger screw speeds (see [11] and references therein), which are the rheological signatures of improved clay exfoliation. However, since CPNC do not follow the Cox-Merz rule [27,31,35], improvement of clay dispersion at lower screw speeds cannot readily be inferred from the flow curves reported in Figure 4. Indeed, it has been reported [25] that nonlinear steady shear properties are less sensitive to the degree of clay dispersion in PDMS than linear rheological properties such as the shear storage and loss moduli [11,12,13,31].

Consequently, the in situ morphological information retrieved from both SALS patterns analysis and turbidimetry was used here to clarify the relationship between the degree of clay dispersion and the magnitude of the flow curves. The clay particle radius *R* was inferred from the light intensity profiles *I*(*q*) extracted from the circular average of the isotropic SALS patterns. To do so, *I*(*q*) curves were fitted with the Debye-Andersen-Brumberger equation which reads [36]:(2)$$I\left(q\right)=\frac{A}{{\left(1+{\left(qR\right)}^{2}\right)}^{2}}$$
where *A* is a constant and *q* relates to the angle *θ* of the scattered light as $q=\frac{4\pi n_{r}sin\left(\theta /2\right)}{\lambda }$, with *n_r_* the refractive index of the CPNC and *λ* the wavelength of the light source (here a HeNe laser with *λ* = 633 nm). Unfortunately, the small *q* range available in the SALS set-up precludes the determination of the form factor of scattering objects [36]. Intensity profiles thus cannot distinguish light scattering by flake-like rigid tactoids from light scattered by spherical particles. The approximation given by Equation (2) seems fully justified when the clay powder is dispersed into aggregates or individual tactoids [10,14], or when the polymer intercalates in the tactoid galleries. It could bring progressively larger errors when thinner/smaller tactoids are generated during processing, en route to the full exfoliation of clay platelets, which are too small to be detected by SALS. Equation (2) falls under the Rayleigh-Debye-Gans approximation, which permits the extraction of the volume fraction *ϕ* of clay particles with radius *R* that is obtained from the measured turbidity *T* using the following equation [37,38]:(3)$$T\left(R,\phi \right)=6\phi R{\left(\frac{2\pi }{\lambda }\right)}^{2}{\left(\frac{n_{p}}{n_{m}}-1\right)}^{2}$$
where *n_p_* is the refractive index of clay particles (taken as 1.43 [39]) and *n_m_* is the refractive index of PDMS (here 1.38, as given by the manufacturer). Figure 5 presents the morphological information (i.e., *R* and *ϕ*) retrieved from SALS patterns and turbidity values measured at each shear rate applied in the slit channel of the rheo-optical die. Note here that fitting Equation (2) to the data returns an averaged (over the scattering volume) particle radius *R*, and ignores the actual distribution in droplet sizes. Although the occurrence of a bimodal distribution would have a strong impact on the error on *R* values computed from the fitting procedure and would require an additional term in Equation (2) for a proper analysis of the intensity profile, computed errors are less sensitive to, e.g., a broadening of the distribution.

As the CPNC is processed at a higher screw speed, more particles with a smaller size are generated. In contrast, Figure 5a reports a drop in *R* from 0.74 μm at 20 rpm down to 0.6 μm at 80 rpm, Figure 5b shows an increase in *ϕ* from 1.0% at lower shear rates and 20 rpm to 1.4% at larger shear rates and 80 rpm. A shear rate dependence in both *R* and *ϕ* when the lowest screw speed is used is also perceived in Figure 5: *R* decreases with increasing shear rates while *ϕ* increases, but both level-off beyond 20 s^−1^. At higher screw speeds, *R* and *ϕ* do not show any significant shear rate dependence.

The downsizing of the clay tactoids measured in the rheo-optical slit should relate with the thermomechanical history accumulated along the extruder barrel, i.e., with the level of stresses and the time during which they were exerted, and an eventual contribution of the die itself. This seems to be the case when the extruder operates at 20 rpm, as both *R* and *ϕ* depend on the shear rate at the slit.

The average shear rates ${\dot{\gamma }}_{av}$ generated in the extruder channel at different screw speeds *N* can be approximated with the following relationship [40], which is valid for the screw channels working fully-filled, where most of the mixing and morphology evolution should take place:(4)$$\;{\dot{\gamma }}_{av}=\frac{\pi DN}{60h}$$
where *h* is the average channel depth (taken as 1.2 mm). Equation (4) gives an average shear rate of 11 s^−1^ when a screw speed of 20 rpm was used, which was within the lower shear rates generated in the measuring slit. Therefore, a contribution to particle break-up or erosion conducing to smaller *R* and larger *ϕ* at larger$\;\dot{\gamma }$ could develop in the rheo-optical slit. However, at screw speed *N* = 80 rpm, ${\dot{\gamma }}_{av}$ = 45 s^−1^; this is as large as the maximum shear rate generated in the rheo-optical slit. Thus, in this case, the contribution of the measuring slit to particle size reduction should be negligible. These estimates assume that the residence time between the screws tips and the optical window of the measuring channel is shorter than the time needed for particle re-agglomeration, and the absence of any viscous heating, with the associated reduction of stresses. It should also be noted that local shear rates in co-rotating twin-screw extruders vary significantly. While the minimum shear rate is approximately 90% of the average value, the maximum shear rate at the gaps can be 30 times larger than the average value computed by Equation (4) [40].

An increased number of smaller particles due to exfoliation (either by break-up or erosion) generates more interfacial contributions to the CPNC viscosity, which is thus expected to increase with increasing screw speed. Contrarily, clay intercalation by diffusion of polymer chains into the clay galleries should have a lesser effect on viscosity. However, Figure 4 shows that the CPNC viscosity increased with decreasing screw speed. It is important to note that the incorporation of 2.5 wt.% clay corresponds to a volume fraction of 1.44%. The values of *ϕ* reported in Figure 5b represent between 69% (at a lower shear rate and 20 rpm) and 97% (at 80 rpm) of the clay content. Therefore, whereas the viscosity probed in-situ at faster screw speeds originates mostly from the clay particles with the dimensions presented in Figure 5a, in the case of low screw speeds, one should bear in mind that a non-negligible percentage of particles has sizes lower than 100 nm (the cut-off length scale for the present SALS set-up [41]).

In order to confirm this hypothesis, Figure 6 compares the CPNC viscosities measured during extrusion at 20 and 80 rpm with the flow curves computed from an expansion (in shear rate$\;\dot{\gamma }$) of the Einstein equation for a Newtonian diluted suspension of spheres [42,43], which reads:(5)$$\eta \left(\phi ,\;\dot{\gamma }\right)=\eta \left(\dot{\gamma }\right)\left(1+2.5\phi \left(\dot{\gamma }\right)\right)$$
where $\eta \left(\phi ,\;\dot{\gamma }\right)$ is the steady shear viscosity of the CPNC computed from the shear viscosity of the PDMS matrix at the corresponding shear rate (also included in Figure 6) $\eta \left(\dot{\gamma }\right)$, given by Equation (1) using the parameters reported for the fit shown in Figure 3b), and $\phi \left(\dot{\gamma }\right)$ is the clay volume fraction at the corresponding shear rate reported in Figure 5b.

A satisfactory match between the computed and measured shear viscosities is observed in Figure 6 for the CPNC processed with a screw speed of 80 rpm within the whole range of shear rates, whereas a discrepancy between the computed and measured values at lower shear rates is evident for the CPNC processed at 20 rpm. Most probably, at the latter screw speed, particles with a size below 100 nm and representing 31% of the added clay are responsible for the increase in viscosity.

It has been postulated that break-up of particle agglomerates is associated with the development of sufficiently high stresses, this being a relatively fast process, whereas erosion can develop under lower stresses, but requires longer times [44,45,46]. The prevalence of each dispersion route can be predicted based on the magnitude of the fragmentation number *Fa*, which compares the hydrodynamic stress *σ* with the cohesive strength *σ_c_* of primary particles [44,45,47]. Thus, the in-process rheo-optical characterization of clay dispersion in PDMS presented in Figure 4 and Figure 5 seems to indicate that larger screw speeds (and thus higher stresses) promote the break-up of clay particles into smaller ones, whereas lower screw speeds should favor the erosion of the clay tactoids. In both cases, there is a progressive generation of smaller anisotropic objects with large hydrodynamic effects on the CPNC rheology.

Various models are available to predict the kinetics of particle dispersion, and thus they can be used to compare the predicted radii *R* with the values measured at the smallest shear rate applied in the rheo-optical slit. Rwei et al. [48] proposed the following equation assuming erosion:(6)$$\ln R\left(t\right)=\ln R_{0}-C\dot{\gamma_{av}}t$$
where *R*_0_ is the initial particle size of the clay, *C* is a constant and *t* is the average residence time of clay particles in the extruder, which is operated at a screw speed, inducing the corresponding $\dot{\gamma_{av}}$. Collin et al. [49] proposed a relationship valid for large *Fa* numbers and negligible cohesive strength of the clay particles:(7)$$R{\left(t\right)}^{3}=R_{0}^{3}-C_{1}\sigma_{av}\dot{\gamma_{av}}t$$
where *C*_1_ is a constant, and *σ_av_* is the average stress applied on the clay particles in the extruder. Figure 7 displays a graphical representation of Equations (6) and (7) with the values of *R* reported in Figure 5a for the lowest shear rate and with $\dot{\gamma_{av}}$ measured and computed as reported above for the corresponding screw speeds, whereas the values of *t* are presented in Table 1. This calculation is just an approximation since *σ_av_*, and $\dot{\gamma_{av}}$ only develop in zones where the screw channels operate fully filled, while *t* refers to the entire screw (and die) length, but it should still allow to eventually extract interesting correlations (in fact, fully filled channels contribute significantly for the total residence time). The average stress is computed from:(8)$$\sigma_{av}=\eta \left(\dot{\gamma_{av}}\right)\dot{\gamma_{av}}$$
where $\eta \left(\dot{\gamma_{av}}\right)$ is calculated with Equation (1) using $\dot{\gamma }=\dot{\gamma_{av}}$ and the parameters obtained from the fit to the data in Figure 3b.

Linear relationships are seen in Figure 7 when CPNC are processed at lower screw speeds, which correspond to lower average shear rates $\dot{\gamma_{av}}$ and stresses *σ_av_* (see values listed in Table 1). Such linearity shows that both Equations (6) and (7) hold, thus indicating that clay particle erosion takes place. However, the *R*-value measured at the highest screw speed tested (80 rpm) departs significantly from the linear trend, which again suggests a change in the dispersion route, probably from particle erosion to rupture as the screw speed is increased.

As discussed above, it has been proposed to use the fragmentation number *Fa* to estimate the threshold conditions for dispersion, as well as the dispersion route. For example, for silica agglomerates of various densities suspended in low molecular weight polymers with different viscosities, Scurati et al. [44] confirmed that the critical stress for erosion is smaller than that for break-up, and showed that erosion takes place when 2 ≤ *Fa* < 5, while break-up develops quickly when $F_{a}\geq 5$. Assuming that these relationships hold for the organoclay used here, its cohesive strength *σ_c_* can thus be computed from the average and maximum stresses estimated at 20 rpm and *Fa* = 2, which gives 3.3 < *σ_c_* < 26.9 kPa. Since rupture most likely occurred at 80 rpm (as assumed above from the information conveyed in Figure 7), *σ_c_* can also be appraised from the average and maximum stresses calculated at 80 rpm, but now assuming that *Fa* is at least 5. This yields 3.6 < *σ_c_* < 19 kPa, which is quite similar to the other estimate.

The lower values for *σ_c_* are 6 times larger than the cohesive strength measured for silica agglomerates [44], but the upper limits are of the same order of magnitude as the range of stresses needed to peel off a 1 µm clay tactoid into two halves, for tactoids with stacking defects allowing peeling angles larger than 3° [14]. Interestingly, this range is also within the magnitude of the predicted cohesive strength of graphite nanoplates [50].

The inset in Figure 7 compares the WAXD spectra of CNPC extrudates collected during experiments performed at 40 and 80 rpm. These spectra confirm that little or no PDMS intercalation within the clay galleries occurred, as the diffraction peaks in CPNC are essentially similar to that of the clay powder. Evidently, the amount of clay platelets exfoliated or peeled-off from the parent tactoids at the lowest screw speed is too small to give a significant reduction in the peak intensity.

In summary, the set of data presented so far underlines that a different level of clay dispersion is achieved at different screw speeds, but it does not relate simply to a progressively better PDMS intercalation and higher clay exfoliation as screw speed is increased. Rather, an interplay between smaller stress and longer residence time leads to a preferred erosion of clay particles at lower screw speeds, whereas clay particle break-up is more likely to occur at larger screw speeds, as the latter impose larger stresses on the clay tactoids. This result is only in partial agreement with the classical dispersion mechanism, which regulates the need for PDMS diffusion into the clay galleries before achieving clay dispersion.

### 3.3. Effect of Temperature

To reach a high level of exfoliation, it is been recommended first to mix clay and the polymeric matrix in a molten state under mild shearing conditions to facilitate intercalation, i.e., the diffusion of polymer chains into the confined space between the galleries, followed by a high level of shearing for break-up of the tactoids [9,51,52]. Therefore, temperature was used to lower PDMS viscosity in the feed and first mixing zones of the extruder. No heating was used downstream, in order to thicken the PDMS and generate higher stress for break-up.

Figure 8a shows the evolution of the CNPC temperature along the barrel measured for all screw speeds tested by sticking a fast response thermocouple into the material flowing out from the sampling ports. Shear stress profiles along the barrel are reported in Figure 8b.

As the screw speed increases from 20 to 80 rpm, the rise in CPNC temperature in the first part of the extruder declines by nearly 15 °C, due to the associated reduction of the local residence time. Contrariwise, and as expected, heat relaxation along the barrel is less efficient at larger screw speeds. Stress profiles along the barrel are reported in Figure 8b for all screw speeds but 20 rpm, as the latter did not generate sufficient pressure in the rheo-optical die to guarantee a stable output of material. The stresses at each location were computed using Equation (8), taking into account the effect of temperature presented in Figure 8a on the local shear viscosity by substituting *η_0_* in the computation of $\eta \left(\dot{\gamma_{av}}\right)$, with its Arrhenius dependence determined by the fit reported in the inset of Figure 3a. As anticipated, the average shear stress experienced by the CPNC is lower in the first part of the extruder (by approximately 50% with respect to tests conducted with no heating, compare the data in Figure 8b with that in Table 1), and increases downstream. Figure 9 displays the CPNC flow curves and microstructural properties when the barrel was heated.

The flow curves plotted in Figure 9a confirm the effect of increasing screw speed on the reduction of the CPNC shear viscosity already observed in Figure 4 for the lower shear rate regime, but the magnitude of the viscosity drop is smaller. However, the underlying morphological changes are different, as seen in Figure 9b,c, which depict the shear rate dependence of the particles radii and volume fractions (and can be compared with Figure 5a,b. It appears that in the low shear rate regime less particles (smaller *ϕ*) with larger radii *R* are generated at the largest screw speed tested (*N* = 80 rpm). This trend, which is the total opposite to that observed above when no heating was used (see Figure 5), is less evident at 40 and 60 rpm, as these speeds seem to generate similar morphologies because the corresponding flow curves are virtually identical. The effect of barrel heating on the CPNC morphology is more clearly illustrated in Table 2, which compares, for selected shear rates, the *R* and *ϕ* measured for the three screw speeds with and without heating. Data in Table 2 confirms that barrel heating in the initial part of the extruder enables achieving a greater number of smaller clay particles when screw speeds of 40 and 60 rpm are used (except for data measured at 43 s^−1^ with *N* = 60 rpm). No enhancement of clay dispersion with heating is perceived at 80 rpm. Probably, a complex interaction between stress and residence time is again responsible for the distinct results obtained for clay dispersion at 80 rpm, which will be investigated in the following section.

### 3.4. Separating the Effects of Residence Time and Stress on the Level of Clay Dispersion

During extrusion with partial barrel heating, *t* cannot be tuned separately to match the residence times used in the corresponding experiments performed at room temperature. This is illustrated in Figure 10, where the average residence times *t* measured in the two sets of experiments are compared. The figure also plots an average mechanical power, given by the product of the viscosity by the square of the average shear rate. As expected, higher average residence times *t* and lower mechanical power are associated with extrusion runs with heating. In addition, increasing screw speeds (at constant throughput) reduces the average residence time but increases the mechanical work.

Figure 10 also shows that two sets of extrusion trials allow isolating the effects of *t* and $\eta {\dot{\gamma }}^{2}$ on the level of clay dispersion. The experiments performed at *N* = 40 rpm, with and without partial barrel heating, show nearly the same level of $\eta {\dot{\gamma }}^{2}$ (250 kPa·s^−1^) but *t* are significantly different (average residence times of *t* = 318 and *t* = 287 s, respectively). Inversely, the experiments carried out with *N* = 80 rpm yielded similar average residence times (*t* = 263 ± 1 s), but the mechanical power was distinct (512 kPa·s^−1^ and 825 kPa·s^−1^ for partial barrel heating and no heating, respectively). The results of these two sets of experiments in terms of shear viscosity, clay particles radii and volume fraction are compared in Figure 11 and Figure 12.

Increasing the average residence time of the CPNC in the extruder favors dispersion, with the corresponding increase of its shear viscosity (Figure 11). Consistently, the tactoids comprise fewer clay lamellae since the WAXD spectrum of the corresponding extruded CPNC presents a peak with a reduced width (1.53 ± 0.16° against 2.03 ± 0.34°, as computed from Lorentzian fits to the peaks). Thus, longer residence times coupled to a reduced PDMS viscosity favor polymer diffusion into the clay galleries, which then facilitates clay exfoliation.

Figure 12 shows that increasing $\eta {\dot{\gamma }}^{2}$ by a factor of nearly 1.6 has a small effect on clay dispersion, since comparable particle sizes and volume fractions are measured in the two sets of experiments with and without heating, which accordingly exhibit identical flow curves. The data in Figure 7, obtained for experiments performed without heating, suggested that at 80 rpm, dispersion could involve both erosion and break-up. Thus, the similar level of dispersion depicted in Figure 12 (in fact, a close observation showed somewhat smaller *R* and higher *ϕ* for higher$\;\eta {\dot{\gamma }}^{2}$) probably meaning that at *N* = 80 rpm the average residence time in the first part of the extruder is insufficient to allow for significant intercalation, the higher stresses downstream playing a major role in dispersion.

In a recent study, Ferrás et al. [50] reported a quasi-linear relationship between the degree of microscopic dispersion of graphite nanoplates in polypropylene and the product of the average residence time by the stress applied on the particles in a prototype mixer. This product was an attempt to consider the combined effects of stress and residence time on dispersion. A test for such a linear relationship is offered in Figure 13, where the average radii of the particles and the corresponding fraction are plotted as a function of $\sigma_{av}t$. *R* and *ϕ* correspond to the dispersion indicator used by Ferrás et al., namely the ratio *A* of the area of all particles imaged by optical microscopy over the total pictured area. The decay of *R* and the rise in *ϕ* with increasing $\sigma_{av}t$, when extrusion is performed without heating, is consistent with the quasi-linear decrease of *A* with σt underpinned by Ferrás et al. [50], and confirm that the conjugation of stress and residence time favors dispersion. The trend of *R* and *ϕ* when partial barrel heating is used is more difficult to explain. It should be noted that $\sigma_{av}$ varied significantly along the extruder (see Figure 8b), hence even taking the average of four values at different downstream locations to compute it might entail a considerable error. Anyway, it is interesting to note that at constant $\sigma_{av}$ partial barrel heating always yields better dispersion levels (lower *R*, higher *ϕ).* This underlines the contribution of intercalation, i.e., of the diffusion of polymer molecules into the clay galleries, to achieve higher dispersion levels by break-up and/or erosion.

The inset in Figure 13 represents the variation of turbidity *T* with $\sigma_{av}t$. *T* is often used as a measure of dispersion [23,39], but its value results from the contribution of particle size and the number of particles, which evolve in the opposite direction as dispersion proceeds. In the present case, turbidity showed the same trend of *ϕ* and the opposite trend of *R*. This demonstrates that turbidity should be used with caution to assess dispersion levels.

## 4. Conclusions

The in-line rheo-optical investigation of clay dispersion during the twin-screw extrusion of a model CPNC provided innovative data on the relationships between stresses, strains, residence time, and levels of clay dispersion. The experimental approach proposed here enabled us to bypass most of the experimental issues that eventually polluted the data generated in earlier studies, such as thermal degradation, post extrusion rearrangement of the CPNC structure, and simultaneous melting and mixing. As a result, a complex interplay between screw speed and the level of dispersion is demonstrated here, which cannot be rationalized by a simple relationship between larger screw speeds (or mechanical energy) and enhanced clay dispersion.

Particle erosion and break-up contribute simultaneously to dispersion, which implies that similar dispersion qualities could be eventually achieved by the two different mechanisms if sufficient residence time was provided. PDMS intercalation into clay galleries enhances clay erosion or break-up. Consequently, applying larger stresses (e.g., increasing the screw speed) does not simply translate into better clay dispersion, whereas longer residence times do.

Even though a model nanocomposite was studied and a small-scale twin-screw extruder was used, it is expected that the conclusions obtained will remain generally valid for practical/commercial nanocomposites processed in larger machines. Similarly, although the clay concentration in the CPNC was relatively small in order to avoid significant multiple light scattering, a comparable behavior is anticipated from more concentrated CPNCs.

More importantly, the experimental approach presented here, namely the in-line rheo-optical monitoring of twin-screw extrusion, is an ideal testbed for developing new nanocomposites containing other types of fillers and/or setting the most adequate compounding conditions.

Therefore, future work on the topic should follow two directions: (i) obtain rheological and morphological data for the same model system processed using screws with different geometries, thus generating distinct temperature/stress profiles; (ii) test the sensitivity of the in-line rheo-optical system to perturbations in clay concentration and/or operating conditions, in order to assess its potential for quality process monitoring.

## Figures and Tables

**Figure 1 polymers-13-02128-f001:**
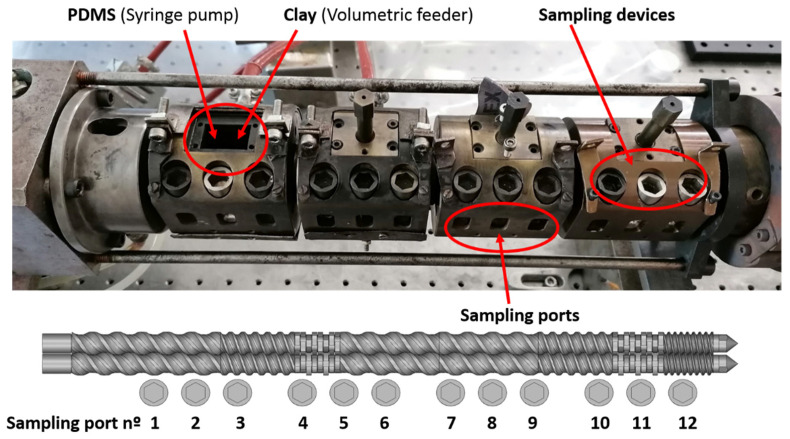
The barrel of the prototype twin-screw extruder. The barrel comprises four temperature-controlled barrel modules, each containing a feed port that can be tapped and three sampling devices, plus a feeding module upstream. PDMS and clay were fed separately at the locations indicated. The screw configuration is 26T78/13T39/8KBL60°/26T52/26T52/13T39/9KBL60°/6.5T26, where T denotes a conveying element (together with the values of pitch and length) and KB refers to kneading disks (right (R) or left (L), and angle).

**Figure 2 polymers-13-02128-f002:**
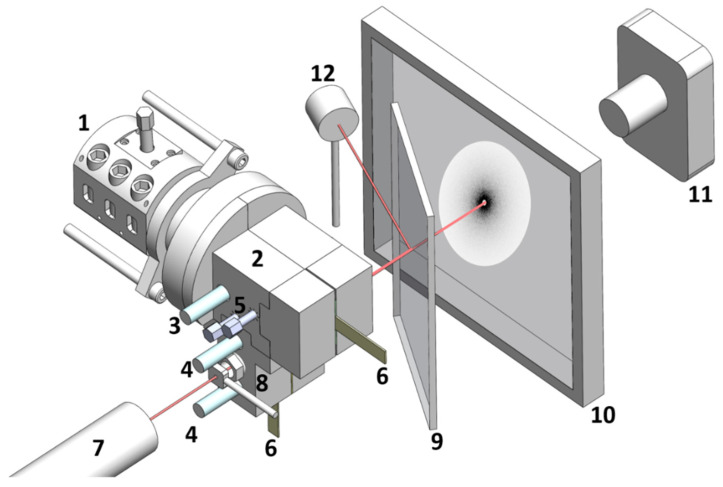
Rheo-optical slit and optical train for the in-situ measurement of flow curves, SALS patterns, and turbidity during extrusion. (1) Extruder; (2) rheo-optical die; (3) pressure transducer at die entry; (4) pressure transducers along the measuring channel; 5) valves for control of material output; (6) extrudate; (7) HeNe laser; (8) pinhole (NRC-ID-0.5, Newport, RI, USA); (9) beam splitter; (10) screen; (11) CDD camera (LU165M-IO, Lumenera, Otawa, Canada); (12) Photodetector (DET-100-002 Hinds Instruments, Hillsboro, OR, USA).

**Figure 3 polymers-13-02128-f003:**
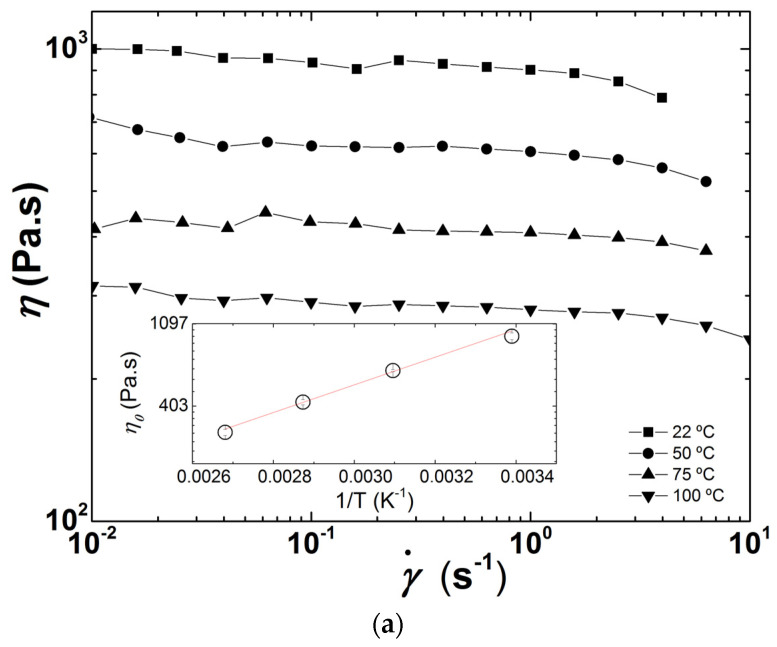
PDMS flow curves: (**a**) measured off-line at different temperatures. The inset is an Arrhenius representation of the thermal dependence of the zero shear viscosity *η*_0_, where the line is an Arrhenius fit to the data; (**b**) measured in-line during two extrusion experiments performed at 24 °C with a screw speed of 40 rpm. The line is a fit of the Cross equation to the two data sets (15 points).

**Figure 4 polymers-13-02128-f004:**
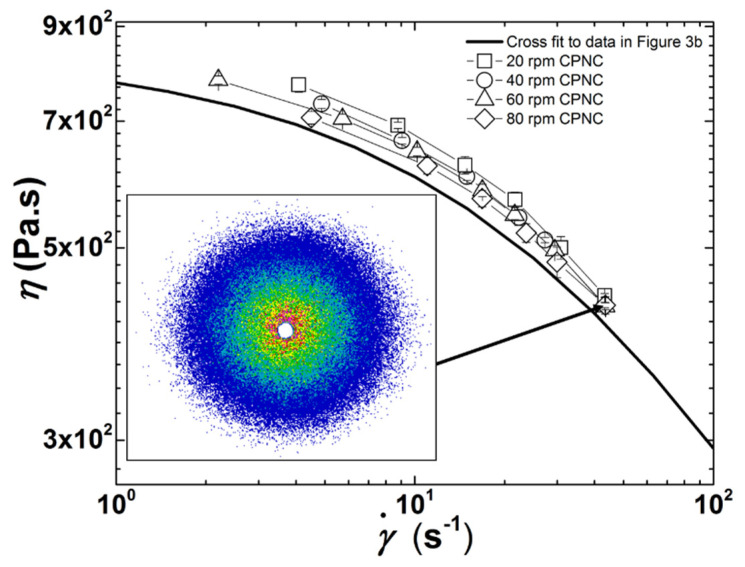
Flow curves measured in-line at different screw speeds. The line is the Cross equation fit to the PDMS flow curve. The inset shows a SALS pattern pictured during extrusion at 80 rpm and with a shear rate of 44 s^−1^ in the rheo-optical slit. Error bars are smaller than the symbols.

**Figure 5 polymers-13-02128-f005:**
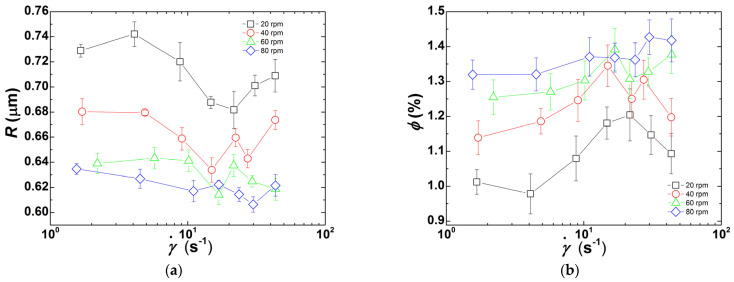
(**a**) Clay particle radius *R* computed from the analysis of SALS patterns recorded in the rheo-optical slit die at the corresponding shear rate $\dot{\gamma }$; (**b**) Volume fraction *ϕ* of clay particles of size *R* measured from turbidimetry performed at the corresponding shear rate $\dot{\gamma }$.

**Figure 6 polymers-13-02128-f006:**
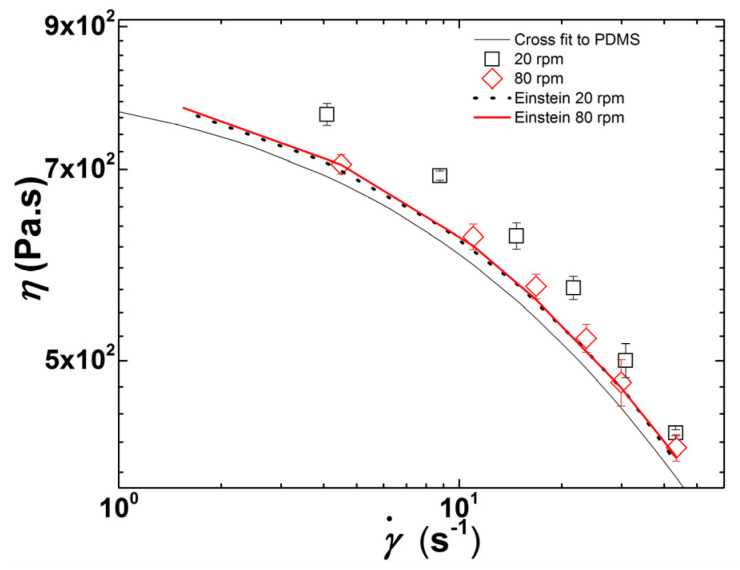
Comparison between the CNPC flow curves measured during extrusion at 20 rpm and 80 rpm, the flow curve of the PDMS matrix computed from the fit of Equation (1) to the data plotted in Figure 3b, and the flow curves computed with Equation (5) using the $\phi \left(\dot{\gamma }\right)$ data reported in Figure 5b.

**Figure 7 polymers-13-02128-f007:**
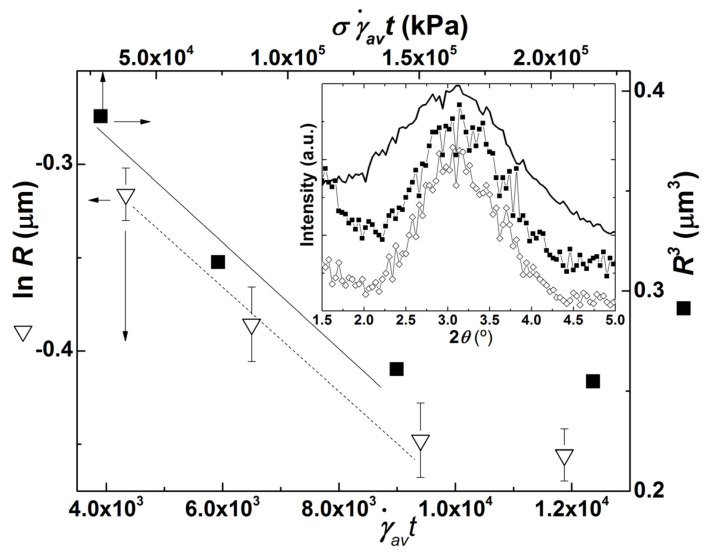
Graphical representations of Equation (6) (open triangles) and Equation (7) (solid squares). Radii *R* of the clay particles obtained at different screw speeds (with corresponding average residence times *t*, average stress *σ_av_* and shear rate $\dot{\gamma_{av}}$) are those reported in Figure 5a at the lowest shear rate for the corresponding screw speed. Solid and dotted lines are guides to the eyes. Inset: WAXD spectra of clay powder (line) and extrudates collected during experiments performed with a screw speed of 40 rpm (solid squares) and 80 rpm (empty diamonds).

**Figure 8 polymers-13-02128-f008:**
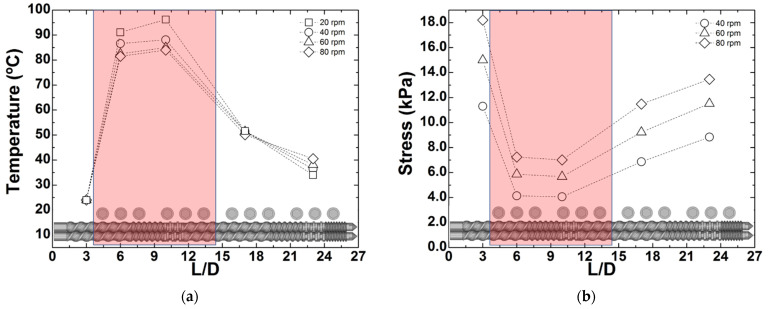
Measured temperature (**a**) and computed shear stress (**b**) profiles along the barrel of the extruder operated at different screw speeds and with barrel heating set to 100 °C at the feed and first mixing zones (shaded region). The experimental error on the temperature readings is of the order of 1 °C, which is smaller than the size of the symbols.

**Figure 9 polymers-13-02128-f009:**
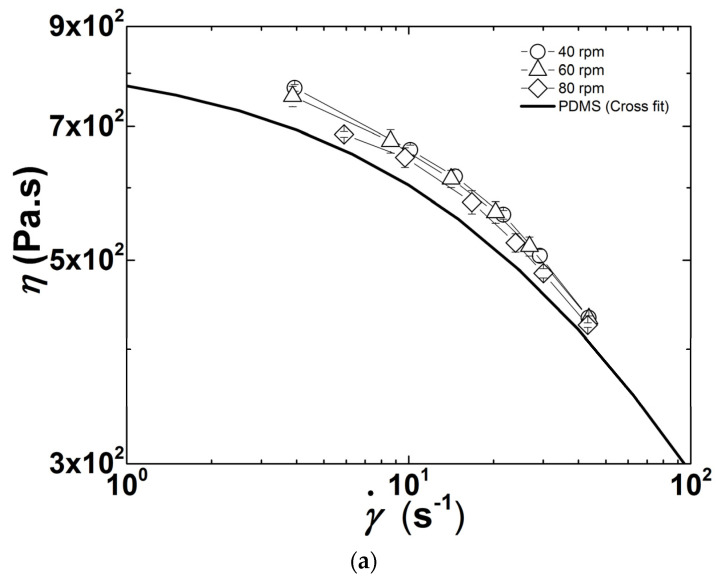
Shear rate ($\dot{\gamma }$) dependence of shear viscosity *η* (**a**), clay particles radii *R* (**b**), and corresponding volume fraction *ϕ* (**c**), recorded in-line for extrusion runs performed at different screw speeds with heating of the feed and first mixing zones.

**Figure 10 polymers-13-02128-f010:**
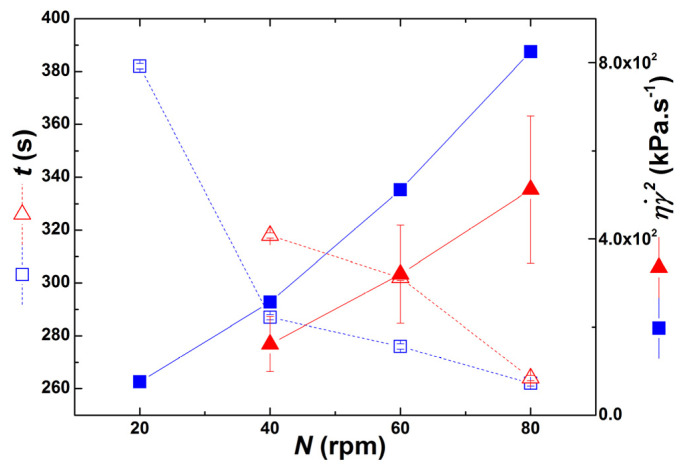
Average residence time *t* (empty symbols) and average mechanical power $\eta {\dot{\gamma }}^{2}$ (solid symbols) for the extrusion trials performed at different screw speeds *N* without (blue squares) and with (red triangles) partial barrel heating. Standard deviations on $\eta {\dot{\gamma }}^{2}$ for the heating experiments are plotted as vertical bars.

**Figure 11 polymers-13-02128-f011:**
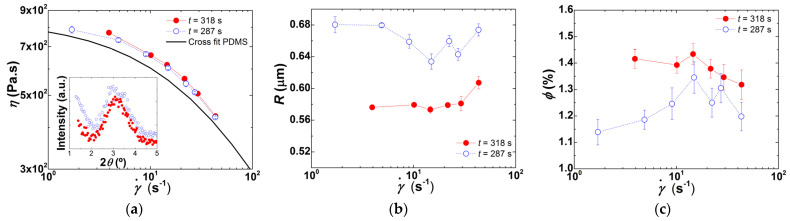
Experiments performed at *N* = 40 rpm, with and without partial barrel heating. Shear rate ($\dot{\gamma }$) dependence of shear viscosity *η* (**a**), clay particles radii *R* (**b**), and corresponding volume fraction *ϕ* (**c**), recorded in-line for extrusion tests performed with (solid symbols) and without (empty symbols) partial barrel heating. Inset to Figure 11a: WAXD spectra of extrudates collected during tests carried out without (empty symbols) and with (solid symbols) partial barrel heating.

**Figure 12 polymers-13-02128-f012:**
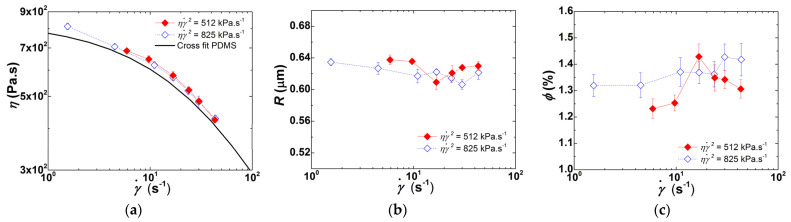
Experiments performed at *N* = 80 rpm, with and without partial barrel heating. Shear rate ($\dot{\gamma }$) dependence of shear viscosity *η* (**a**), clay particles radii *R* (**b**), and corresponding volume fraction *ϕ* (**c**), recorded in-line for extrusion tests performed with (solid symbols) and without (empty symbols) partial barrel heating.

**Figure 13 polymers-13-02128-f013:**
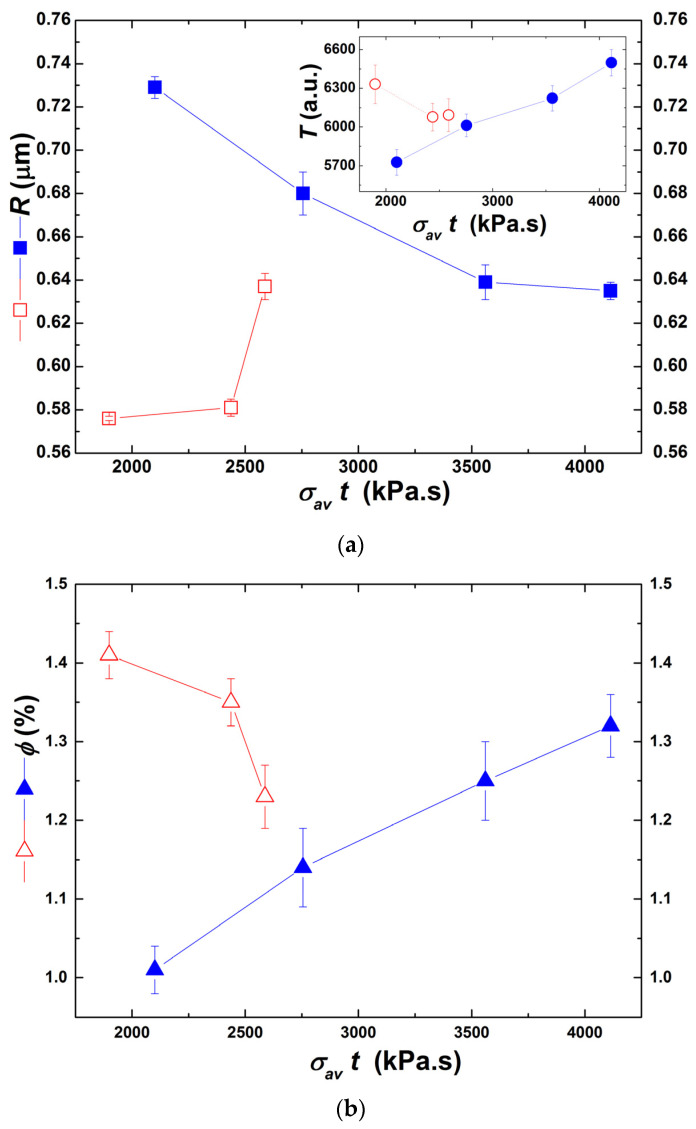
Relationship between clay particles radii *R* (**a**), and corresponding volume fraction *ϕ* (**b**), and the product of the stress by the residence time ($\sigma_{av}t$) for extrusion trials carried out with no heating (solid blue symbols) and with partial barrel heating (empty red symbols). Inset in (**a**): turbidity *T* measured in-line during extrusion trials carried out with the corresponding product of stress by residence time.

**Table 1 polymers-13-02128-t001:** Measured average residence time *t*, computed average shear rate $\dot{\gamma_{av}}$, average stress *σ_av_*, and maximum stress *σ_max_* (computed from the maximum shear rate taken as $\dot{30\gamma_{av}}$ ) for each corresponding screw speed *N*.

*N* (rpm)	*t* (s) ^a^	$\dot{\gamma_{av}}\;(s^{-1}){\;}^{b}$	*σ_av_* (Pa) ^c^	*σ_max_* (Pa) ^c^
20	382.5	11	6671	53,917
40	286.6	23	11,313	72,458
60	276.4	34	15,037	85,102
80	261.8	45	18,193	94,988

^a^ Experimental error is 0.5 s. ^b^ Computed from Equation (4). ^c^ Computed from Equation (8).

**Table 2 polymers-13-02128-t002:** Clay particles radii *R* and corresponding volume fraction *ϕ* measured in the rheo-optical die, at three constant shear rates (4, 10, and 43 s^−1^) during extrusion runs with different screw speeds *N* (40, 60, and 80 rpm), performed without and with (partial) barrel heating. Errors in *R* are computed from the fit of Equation (2) to the light intensity profiles. Errors in *ϕ* result from the cumulative weighted errors originating from the averaging of the steady turbidity signals over 10 s and from the error in *R*.

***N*** ** = 40 rpm**	$\dot{\gamma }$ ** = 4 s^−1^**		$\dot{\gamma }$ ** = 10 s^−1^**		$\dot{\gamma }$ ** = 43 s^−1^**	
	***R*** **(μm)**	***ϕ*** **(%)**	***R*** **(μm)**	***ϕ*** **(%)**	***R*** **(μm)**	***ϕ*** **(%)**
No heating	0.68 ± 0.01	1.18 ± 0.05	0.66 ± 0.01	1.24 ± 0.06	0.67 ± 0.01	1.20 ± 0.05
Heating	0.58 ± 0.01	1.42 ± 0.04	0.58 ± 0.01	1.39 ± 0.03	0.61 ± 0.01	1.32 ± 0.05
***N*** ** = 60 rpm**	$\dot{\gamma }$ ** = 4 s^−1^**		$\dot{\gamma }$ ** = 10 s^−1^**		$\dot{\gamma }$ ** = 43 s^−1^**	
	***R*** **(** **μ** **m)**	***ϕ*** **(%)**	***R*** **(** **μ** **m)**	***ϕ*** **(%)**	***R*** **(** **μ** **m)**	***ϕ*** **(%)**
No heating	0.64 ± 0.01	1.26 ± 0.05	0.64 ± 0.01	1.30 ± 0.06	0.62 ± 0.01	1.38 ± 0.05
Heating	0.58 ± 0.01	1.35 ± 0.03	0.58 ± 0.01	1.45 ± 0.05	0.63 ± 0.01	1.30 ± 0.06
***N*** ** = 80 rpm**	$\dot{\gamma }$ ** = 4 s^−1^**		$\dot{\gamma }$ ** = 10 s^−1^**		$\dot{\gamma }$ ** = 43 s^−1^**	
	***R*** **(** **μ** **m)**	***ϕ*** **(%)**	***R*** **(** **μ** **m)**	***ϕ*** **(%)**	***R*** **(** **μ** **m)**	***ϕ*** **(%)**
No heating	0.63 ± 0.01	1.32 ± 0.04	0.62 ± 0.01	1.27 ± 0.05	0.62 ± 0.01	1.42 ± 0.06
Heating	0.63 ± 0.01	1.23 ± 0.04	0.63 ± 0.01	1.25 ± 0.03	0.63 ± 0.01	1.31 ± 0.03

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
