# Peer review of "In-Line Rheo-Optical Investigation of the Dispersion of Organoclay in a Polymer Matrix during Twin-Screw Compounding"

_polymers, 2021, doi:10.3390/polym13132128_

Round 1
Reviewer 1 Report
The manuscript ,,In-line rheo-optical investigation of the dispersion of organoclay in a polymer matrix during twin-screw compounding,, contains many interesting findings and results. It is also well structured. However, the manuscript still needs to be improved. A major revision is required.
My advice, recommendations and comments are listed below.
Please be sure that your manuscript thoroughly establishes how this work is fundamentally novel. Specific comparisons should be made to previously published materials that have a similar purpose. Please present a strong case for how this work is a major advance. This needs to be done in the manuscript itself, not just in the response to review comments. This is a very important point in terms of which I will further consider the manuscript.
Please be sure that your abstract and your Conclusions section not only summarize the key findings of your work but also explain the specific ways in which this work fundamentally advances the field relative to prior literature.
The significance of this study should be more emphasize in the introduction. Take a look at this paper that may help you. https://link.springer.com/article/10.1007/s00397-007-0243-2
Line 30-32, organoclay: This issue has been addressed in great detail by this very important paper and therefore the authors are encouraged to add it here as a reference. https://www.sciencedirect.com/science/article/abs/pii/S0169131719301413
Line 32: Concretize and specify practical applications. Indicate appropriate specific cases.
Line 40: ,,clay interlayer spacing (intercalation),, This is derived from the value of d001 so-called basal spacing. This was also investigated by Vaia and the co-authors. Based on the XRD, a different arrangement can be observed.
Pure clay, for example sodium montmorillonite, has a d001 of about 1.25 nm, and after intercalation with organic matter, the so-called expansion of the interlayer space is stretched and depends on the size of the intercalated molecule.
This also has a significant effect on the properties of individual polymer nanocomposites. The hydrophilic surface of the clay can become hydrophobic by exchanging hydrated inorganic cations for organic ones, and these materials are called organoclay and are suitable for a wide range of applications. You should develop it in the manuscript and discuss it.
Line 127: Specify the type of montmorillonite. It was the sodium or calcium form. In the manuscript should indicate the chemical composition (individual oxides) or the resulting structural formula.
Line 201: Why this range of angles?
Line 218: Please enlarge this image there are poorly visible descriptions on it.
Line 225: Improve the notation and interpretation of equations with explanations throughout the manuscript.
Line 292: Distinguish the curves by color, so the differences will be more visible.
Line 466: It is necessary to enlarge these figures.
Line 489: What were the measurement deviations?
Line 575: Indicate the possible risks of such research. Add your recommendations for future research.
Line 607: Make sure the references are added correctly according to the journal's instructions.
Author Response
COMMENT 1: Please be sure that your manuscript thoroughly establishes how this work is fundamentally novel. Specific comparisons should be made to previously published materials that have a similar purpose. Please present a strong case for how this work is a major advance. This needs to be done in the manuscript itself, not just in the response to review comments. This is a very important point in terms of which I will further consider the manuscript.
ANSWER 1: In the current version of the manuscript, the introduction contains the following text concerning the novelty of the work:
“There is thus ample room for additional studies where thermal degradation and post-extrusion characterization of CNPC are avoided, while controlling stress and residence time. The present work brings three innovative approaches to the study of the dispersion mechanisms of polymer-organoclay nanocomposites:
- a rheo-optical die is coupled to a small-scale prototype twin screw extruder. The in- situ characterization of the CPNC morphology and viscosity bypasses all issues related with a possible post-extrusion structural rearrangement;
- a model CPNC, namely an organo-modified montmorillonite in a polydimethylsiloxane (PDMS) is used. This CPNC can be processed at ambient or low temperatures (up to 100 ºC, which is well below the onset for PDMS or organo-clay thermal degradation [21]);
- processing this CPNC avoids the need to superimpose polymer melting with mixing.”
Following the guidance of the reviewer, we tried to present a stronger case concerning the novelty of the work as follows:
“It is therefore interesting to study the dispersion mechanisms of polymer-clay nanocomposites adopting methodologies that avoid possible thermal degradation effects and post-extrusion characterization of the materials, whilst controlling stress and residence time. In order to accomplish this, the present work adopts simultaneously three innovative approaches:
- a rheo-optical die is coupled to a small-scale prototype twin screw extruder. The in- situ characterization of the CPNC morphology and viscosity bypasses all issues related with a possible post-extrusion structural rearrangement;
- a model CPNC, namely an organo-modified montmorillonite in a polydimethylsiloxane (PDMS) is used. This CPNC can be processed at ambient or low temperatures (up to 100 ºC, which is well below the onset for PDMS or organo-clay thermal degradation [24]);
- processing this CPNC also avoids the need to superimpose polymer melting with mixing.
Rheo-optical methods have been used previously both off-line [23] and in-line [20] to study the dispersion of organoclay in Poly(lactic acid) (PLA) matrices. PDMS-clay nanocomposites have been previously characterized, but were prepared by solution intercalation methods [28,32,52]. To the authors knowledge, this is the first time that PDMS-organoclay nanocomposites are prepared by melt mixing, and that their dispersion is monitored by in-process rheo-optical methods. Moreover, the incorporation of clays in a melt rather than the joint feeding of solid premixes is rarely reported in the literature.”.
With this new added reference: [52] Burnside, D., Giannelis, V.P. Synthesis and properties of new poly(dimethylsiloxane) nanocomposites. Chem. Mater. 1995, 7, 1597-1600.
COMMENT 2: Please be sure that your abstract and your Conclusions section not only summarize the key findings of your work but also explain the specific ways in which this work fundamentally advances the field relative to prior literature.
ANSWER 2: We checked that lines 10-12 and lines 21-23 in the abstract explain how and what specific advances in the field have been done in this work. In the conclusion, we also checked that lines 578-581 explain how the fundamental advances with respect to the literature have been achieved. The 2 main advances are clearly stated in lines 582-583 which read “…[results]… cannot be rationalized by a simple relationship between larger screw speeds (or mechanical energy) and enhanced clay dispersion”, and lines 588-589 which read “Consequently, applying larger stresses (e.g. by increasing the screw speed) does not simply translate into better clay dispersion, whereas longer residence times does”.
COMMENT 3: The significance of this study should be more emphasize in the introduction. Take a look at this paper that may help you. https://link.springer.com/article/10.1007/s00397-007-0243-2.
ANSWER 3: This paper is actually cited ([ref.48]) in the original manuscript reviewed. We added the following sentence in the introduction to emphasize more a specific significant aspect of the study, which was pointed out by the paper mentioned by the reviewer (now ref. [23]): “Therefore, efforts have been made to identify relationships between the degree of dispersion assessed by optical means (e.g. light attenuation) and the rheological properties of CPNC [11,20,23], since both experimental techniques can be performed in-situ during extrusion [20]”.
We also believe that the various modifications to the text presented above address satisfactorily this comment.
COMMENT 4: Line 30-32, organoclay: This issue has been addressed in great detail by this very important paper and therefore the authors are encouraged to add it here as a reference. https://www.sciencedirect.com/science/article/abs/pii/S0169131719301413
ANSWER 4: this paper is now cited in the list of references (now [6])
COMMENT 5: Line 32: Concretize and specify practical applications. Indicate appropriate specific cases.
ANSWER 5: we inserted the following sentence: “ ….polymer nanocomposites (CPNC) [2-6]. These materials have found increasing practical application in barrier packaging (films and bottles), in flame retardant electrical cables, as anticorrosive coatings on metals, in rubber automotive compounds, among many others [7]. Nevertheless, despite of the significant body of work …”.
With the following added reference [7]: F Ali, H Ullah, Z Ali, F Rahim, F Khan, Z U Rehman, Polymer-clay Nanocomposites, Preparations and Current Applications: A Review, Current Nanomaterials, 1 (2016) 83-95.
COMMENT 6: Line 40: ,,clay interlayer spacing (intercalation),, This is derived from the value of d001 so-called basal spacing. This was also investigated by Vaia and the co-authors. Based on the XRD, a different arrangement can be observed. Pure clay, for example sodium montmorillonite, has a d001 of about 1.25 nm, and after intercalation with organic matter, the so-called expansion of the interlayer space is stretched and depends on the size of the intercalated molecule. This also has a significant effect on the properties of individual polymer nanocomposites. The hydrophilic surface of the clay can become hydrophobic by exchanging hydrated inorganic cations for organic ones, and these materials are called organoclay and are suitable for a wide range of applications. You should develop it in the manuscript and discuss it.
ANSWER 6: to address this comment we added the following lines after line 40: “Clay modification by exchanging hydrated inorganic cations for organic ones bearing aliphatic chains, which results in the production of so-called organoclay, is central to melt intercalation and the delivery of CPNC for a wide range of applications [2-6]. More important than the expansion of the interlayer space which facilitates the diffusion of polymer chains inside the clay interlayer spacing, clay modification lowers the energy of adhesion between clay lamellae below the energy of polymer chain scission (breakup of C-C covalent bond) [9]. Coming back to the dispersion mechanism, large stresses first……”.
COMMENT 7: Line 127: Specify the type of montmorillonite. It was the sodium or calcium form. In the manuscript should indicate the chemical composition (individual oxides) or the resulting structural formula.
ANSWER 7: The information disclosed by the manufacturer of this commercial product is the one reproduced in the manuscript. We however added in the revised manuscript that the starting material before organo-modification is the sodium form of montmorillonite. But without further data from the manufacturer we cannot indicate the contents in individual oxides. We note here that none of the papers cited in this manuscript, and focusing on the topic of the present study, reports the chemical compositions or the resulting structural formula of the employed organically-modified Montmorillonite.
COMMENT 8: Line 201: Why this range of angles?
ANSWER 8: We aimed at identifying a possible change in the d001 basal spacing of the organoclay in relation with the processing parameters. A line is now added in the revised paper to justify the selected range of angles.
COMMENT 9: Line 218: Please enlarge this image there are poorly visible descriptions on it.
ANSWER 9: Done.
COMMENT 10: Line 225: Improve the notation and interpretation of equations with explanations throughout the manuscript.
ANSWER 10: Done.
COMMENT 11: Line 292: Distinguish the curves by color, so the differences will be more visible.
ANSWER 11: Done.
COMMENT 12: Line 466: It is necessary to enlarge these figures.
ANSWER 12: Done.
COMMENT 13: Line 489: What were the measurement deviations?
ANSWER 13: The following line is now added to the table’s caption to address this question: “Errors in R are computed from the fit of equation (2) to the light intensity profiles. Errors in Φ result from the cumulative weighted errors originating from the averaging of the steady turbidity signals over 10 seconds and from the error in R.”
COMMENT 14: Line 575: Indicate the possible risks of such research. Add your recommendations for future research.
ANSWER 14: The following sentences were added to the conclusion:
“Even though a model nanocomposite was studied, and a small-scale twin screw extruder was used, it is expected that the conclusions obtained will remain generally valid for practical/commercial nanocomposites processed in larger machines. Similarly, although the clay concentration in the CPNC was relatively small in order to avoid significant multiple light scattering, a comparable behaviour is anticipated from more concentrated CPNCs.”.
“Therefore, future work on the topic should follow two directions: i) obtain rheological and morphological data for the same model system processed using screws with different geometries, thus generating distinct temperature/stress profiles; ii) test the sensitivity of the in-line rheo-optical system to perturbations in clay concentration and/or operating conditions, in order to assess its potential for quality process monitoring.”.
COMMENT 15: Line 607: Make sure the references are added correctly according to the journal's instructions.
ANSWER 15: References have been checked thoroughly.
Reviewer 2 Report
The manuscript by these authors is an interesting piece proposing an innovative approach to the study of the dispersion mechanisms of polymer-organoclay nanocomposites by using PDMS. The manuscript was well written and scientifically sound. In my opinion, the results obtained by these authors are worthy of publication. I have just some suggestions for the authors. Considering the use, by the authors, of PDMS as the matrix in this work probably a reader aspects more information about the use of this polymer in this framework or a brief description of the state of the art of its employment. Finally, I would suggest simplifying figure 13, which can be less readable for the readers. In the attached .pdf some minor typos to be corrected.

Author Response
COMMENTS: I have just some suggestions for the authors. Considering the use, by the authors, of PDMS as the matrix in this work probably a reader aspects more information about the use of this polymer in this framework or a brief description of the state of the art of its employment. Finally, I would suggest simplifying figure 13, which can be less readable for the readers. In the attached .pdf some minor typos to be corrected..
ANSWER: The following lines are now added in the introduction to contextualize the use of PDMS as matrix for CPNC and for colloidal suspensions:
“Indeed, PDMS with relatively low molecular mass has been used in the CPNC literature [28,32] as the rheological study of polymer dynamics in the presence of clay is readily amenable at room temperature. Similarly, PDMS has been used to study flow-induced structures in colloidal suspensions (see e.g. [51] and references therein);”.
With a new added reference: [51] Vermant, J. Large-scale structures in sheared colloidal suspensions. Curr. Opin. Colloid In. 2001, 6, 489-495..
"PDMS-clay nanocomposites have been previously characterized, but were prepared by solution intercalation methods [28,32,52]. To the authors knowledge, this is the first time that PDMS-organoclay nanocomposites are prepared by melt mixing, and that their dispersion is monitored by in-process rheo-optical methods. Moreover, the incorporation of clays in a melt rather than the joint feeding of solid premixes is rarely reported in the literature.”.
With this new added reference: [52] Burnside, D., Giannelis, V.P. Synthesis and properties of new poly(dimethylsiloxane) nanocomposites. Chem. Mater. 1995, 7, 1597-1600.
Figure 13 is now modified, presenting the data in two different plots: Fig. 13a and Fig. 13b.
Typos signaled by the reviewer are now corrected.
Reviewer 3 Report
General comments
When reading the paper for the first time, I was quite delighted. However, on rereading a second time, I found quite some parts where I have to ask for clarification, as detailed below, before the paper can be published.
The topic is as old as important and therefore, experimental contributions to this field are highly welcome! The set-up you use is nicely downscaled for studying smaller quantities of samples.
Is “tactoid” the correct term for the small clay particles? I didn’t know it and when looking it up, I found it has to do with liquid crystalline phases.
In your experiments with the rheo-optical slit die, you did measurements with different shear rates. When performing such an experiment at constant screw speed, the material at the inlet to the die should always be the same at the same feed rate. So all you do for changing the shear rate is redistributing your constant flow between the measuring channel and the waste channel straight on. As I see it, changes to the particle size could only occur at the die inlet, since you certainly have a cross section reduction and therefore an elongational component in your flow. But this is always the same at the same screw speed independent of shear rate. Therefore, I would expect that you see neither a change in R as f(rate) nor phi as f(rate). But then, since you still measure effects, they would come from the die and not from the extrusion process you want to study. Can you comment on this?
One headache I have is with your results on R and phi. You indicate those numbers as if they were monodisperse quantities. But effectively, you have a particle size distribution. What does your setup record, if the mean R is the same, but the distribution broadens? From you optical patterns, you will always evaluate an averaged quantity. This topic is not dealt with at all in the paper.
Details
Fig. 4: In the legend, it should read Fig. 3b instead of 2b. To the right of the four rpms, I would write CPNC for clarity.
To me, the shift of the viscosity curves looks strongly as an effect of viscous heating. Did you always measure the temperature of your sample after the slit die?
Line 297: Delete one in
Equ. 4: You have a complex screw geometry with forward, backward and kneading elements. I wonder how meaningful such an averaged shear rate is?
Line 331: Fig. 5a instead of 4a
Line 403ff: The calculation of the cohesive strength stand on soft ground… The uncertainty is quite large. Do you need the cohesive strength in the following?
Line 445: Shear stress, not tress
Fig. 8: From your Fig. 1, I see no active cooling to your extruder. I would therefore expect that the elements down the line, incl. the die, will slowly heat up with time. Did you verify e.g. during recording of Fig. 9a, that the temperature in the slit die remains constant?
Table 2: Can you detail how you determined the uncertainties?
Line 495: The motor power consumption should be proportional to the average mechanical power. Did you measure it?
Ref. 37: Eine neue Bestimmung der Moleküldimension
Ref. 38: Berichtigung zu meiner Arbeit
Round 2
Reviewer 1 Report
The manuscript ,,In-line rheo-optical investigation of the dispersion of organoclay in a polymer matrix during twin-screw compounding,, has been significantly improved. It can now be accepted.